# Speaking Up: Veterinary Ethical Responsibilities and Animal Welfare Issues in Everyday Practice

**DOI:** 10.3390/ani8010015

**Published:** 2018-01-22

**Authors:** Elein Hernandez, Anne Fawcett, Emily Brouwer, Jeff Rau, Patricia V. Turner

**Affiliations:** 1Department of Pathobiology, University of Guelph, Guelph, ON N1G 2W1, Canada; elein@uoguelph.ca (E.H.); ebrouwer@uoguelph.ca (E.B.); 2Sydney School of Veterinary Science, University of Sydney, Camperdown, NSW 2006, Australia; anne.fawcett@sydney.edu.au; 3Department of Population Medicine, University of Guelph, Guelph, ON N1G 2W1, Canada; jarau@uoguelph.ca

**Keywords:** ethics, advocacy, animal welfare, veterinary practice

## Abstract

**Simple Summary:**

Veterinarians have an ethical obligation to provide good care for the animals that they see in practice. However, at times, there may be conflicts between the interests of animal caregivers or owners, the interests of veterinarians and the interests of animals. We provide an overview of why and how veterinary ethics is taught to veterinary students, as well as providing a context for thinking about veterinary ethical challenges and animal welfare issues. We argue that veterinarians are ethically obliged to speak up and ask questions when problems arise or are seen and provide a series of clinical case examples in which there is scope for veterinarians to improve animal welfare by ‘speaking up’.

**Abstract:**

Although expectations for appropriate animal care are present in most developed countries, significant animal welfare challenges continue to be seen on a regular basis in all areas of veterinary practice. Veterinary ethics is a relatively new area of educational focus but is thought to be critically important in helping veterinarians formulate their approach to clinical case management and in determining the overall acceptability of practices towards animals. An overview is provided of how veterinary ethics are taught and how common ethical frameworks and approaches are employed—along with legislation, guidelines and codes of professional conduct—to address animal welfare issues. Insufficiently mature ethical reasoning or a lack of veterinary ethical sensitivity can lead to an inability or difficulty in speaking up about concerns with clients and ultimately, failure in their duty of care to animals, leading to poor animal welfare outcomes. A number of examples are provided to illustrate this point. Ensuring that robust ethical frameworks are employed will ultimately help veterinarians to “speak up” to address animal welfare concerns and prevent future harms.

## 1. Introduction

Expectations for appropriate animal management exist in most developed countries and provide the scaffolding supporting societal expectations for veterinary care of animals. Although veterinarians are assumed to be working in the best interests of animals at all times, in reality, this may depend upon the willingness and ability of any given veterinarian to engage in ethical reasoning and openly question accepted practices that occur routinely with animals or that they may be asked to do. In this paper, we explore the importance of strong ethical training of veterinarians as a means of ensuring good animal welfare in a given society. We also present a series of case examples in which veterinary advocacy and ethical decision-making may or may not have occurred to demonstrate common ethical issues in veterinary practice.

## 2. Ethics in Veterinary Education

### 2.1. Why Is Ethics Taught?

Veterinary ethics teaching and application in practice have changed considerably over the past decade. When it was first formally taught, veterinary ethics dealt mostly with aspects of professionalism, for example, how to refer cases, issues around steering and advertising—what Rollin called matters of “professional etiquette” [1]. Ethical questions around euthanasia decision-making (for example, is it ever acceptable to euthanize a healthy animal?) were not examined [2].

Since then, the teaching of veterinary ethics has expanded. Ethics is now included in most veterinary curricula, not least because it is an expected Day One Competency for veterinary graduates. For example, in its Day One Competencies, the UK’s Royal College of Veterinary Surgeons states that new graduates should be able to “understand the ethical and legal responsibilities of the veterinary surgeon in relation to patients, clients, society and the environment” and have underpinning knowledge and understanding of “the ethical framework within which veterinary surgeons should work, including important ethical theories that inform decision-making in professional and animal welfare-related ethics” [3]. 

There is scope for further development and refinement of veterinary ethics teaching. A joint report of the Federation of Veterinarians of Europe (FVE) and the European Association of Establishments for Veterinary Education (EAEVE) called for more uniform, comprehensive teaching of animal welfare, ethics and law across veterinary schools, stating that “one cannot be a good clinician without being aware of the ethical issues in decision-making in practice” [4]. European Directive 2005/36/EC states that ethics is a core subject of veterinary education, without providing clear competencies for students to achieve in this area [5]. In its revised Principles of Veterinary Medical Ethics, the Canadian Veterinary Medical Association called for veterinary schools to “stress the teaching of ethical and value issues as part of the professional curriculum for all veterinary students” and for the National Board of Veterinary Medical Examiners to “prepare and include questions regarding professional ethics in the North American Veterinary Licensing Examination (NAVLE)” [6]. 

The North American Veterinary Medical Education Consortium (NAVMEC) identified ethics as a core component of leadership, as well as an emerging area of concern for veterinarians, stating that “(veterinarians) are committed to the health and welfare of animals and the protection of human health through ethical practice, professional self-regulation, legal compliance and high personal standards of behaviour and practice. They are guided by a code of ethics and law and a commitment to professional competence, appropriate attitudes and behaviour, integrity, personal well-being and the public good” [7]. NAVMEC has called for colleges of veterinary medicine to create and update course materials on ethics and leadership for use in and sharing among, veterinary schools. The report also identifies the need for “increasing awareness on ethical issues, including genetic modification” [7]. Similarly, the World Organisation for Animal Health (OIE) recommends that veterinary education establishments “… teach ethics and value issues to promote high standards of conduct and maintain the integrity of the profession” [8]. The OIE states that Day 1 veterinary graduates should “understand and apply high standards of veterinary medical ethics in carrying out day-to-day duties” and “provide leadership to society on ethical considerations involved in the use and care of animals by humans” [8]. In a study of veterinary educators examining why ethics was taught, four major themes emerged: ethical awareness, ethical knowledge, ethical skills and developing individual and professional qualities [9]. Ethical skills, including ethical reasoning and reflection, value-aware communication skills and informed decision-making skills may be taught to help reduce moral stress [10].

### 2.2. How Is Veterinary Ethics Taught?

In a discussion of how ethics should be taught to veterinary students (alongside animal welfare science and law), Main and colleagues suggest exposing students to a range of ethical frameworks, including utilitarianism, deontology, rights-based theories, virtue ethics, principle-based ethics and social justice, as well as examination of value systems, alternative views, conflict resolution and decision-making processes [11].

Ethical reasoning is not simply learned by a process of repeated exposure to ethical issues [12]. It remains unknown to what extent the teaching of ethics to veterinary students enables them to minimise or avoid moral stress. There is no accepted gold standard for veterinary ethics education and curricula vary, with ethics taught as a standalone subject in some programs and integrated into other subjects in other programs [13,14].

In a review of published European veterinary curricula, the amount of ethics teaching was variable, as was its contextual framing and where it appeared in the curriculum [4]. An in-depth study of ethics teaching was conducted in three European veterinary schools (Copenhagen, Lisbon and Nottingham). Prominent topics taught were classified under four overarching concepts: theories and concepts (including ethical frameworks and approaches), laws and regulations (including codes of conduct), animal welfare science and professionalism [15]. All three schools taught students one or more ethical theories or frameworks to aid decision making. Similar variability in ethics instruction has been noted in North American veterinary colleges. In a 2011 survey of veterinary colleges in Canada, the USA and the Caribbean, only 62% of responding colleges (13 of 21) indicated that ethics was a core component of the curriculum and a mean of 15.5 h of ethics instruction occurred over the curriculum. Further, only 33% (7 of 21) of colleges indicated that students were formally assessed for ethical knowledge and decision-making [13].

Animal welfare and ethics scholars from eight Australasian veterinary schools developed the One Welfare Portal (http://onewelfare.cve.edu.au/), a shared online curriculum resource incorporating a range of interactive features, including case-based scenarios with guided ethical discussion [16]. The resource includes eight subsections, agreed upon by animal welfare and ethics educators on animal welfare science, ethics, companion animals, production animals, wild animals, animals used in research and teaching, animal use within sport, recreation and display and aquatic animals [13,14,15,16]. It incorporates a virtual online debating platform to facilitate student discussion of potentially polarising topics [17]. 

The global charity World Animal Protection developed a welfare and ethics syllabus and teaching resource, designed to be used globally [18]. This resource incorporates discussion of five ethical frameworks or approaches: contractarianism, utilitarianism, deontology, ethics of care and respect for nature [19]. Other online tools include www.aedilemma.net, an interactive learning tool that allows users to determine whether their ethical reasoning is most consistent with contractarian, utilitarian, relational, animal rights or respect for nature ethical frameworks and approaches [20]. It is not known to what extent such online resources are formally embedded into veterinary school curricula. 

Key ethics textbooks, as well as regular columns such as *In Practice’s* Everyday Ethics column and the ethical question of the month in the *Canadian Veterinary Journal,* tend to apply ethical frameworks and approaches to case-based scenarios and are useful ethics training tools for veterinary students and veterinarians alike [1,21,22].

One of the challenges of providing veterinary students with multiple frameworks in a didactic setting is that they may not have time to become properly acquainted with the strengths and limitations of each approach. Additionally, by critiquing every framework and approach, students may develop the impression that all have flaws and that ethics is therefore no more than opinion, leading to disenchantment [10]. Ethical frameworks should not be looked at as being in competition, rather they are complementary [14]. A comparison of the relative strengths and weaknesses of commonly taught ethical frameworks with an interpretation of the veterinarian’s responsibility to “speak up” are presented in Table 1.

Extracurricular activities, such as participation in the annual Intercollegiate Animal Welfare Judging and Assessment Contest, open to veterinary students and licensed veterinarians, may help to strengthen student and practitioner ethics and welfare vocabulary and reasoning skills [25]. Participants have the opportunity to assess the welfare of animals in different contexts and settings, including live animal scenarios, weigh evidence and develop scientifically-based evaluations. This requires veterinary students and practitioners to integrate science-based knowledge about animal husbandry and preferences with ethical values [25].

## 3. Understanding Veterinary Ethical Challenges and Animal Welfare Issues

Veterinary oaths and professional codes of conduct highlight the obligations of veterinarians to animals, clients, colleagues, the wider community, and, increasingly, themselves. For example, the Canadian Veterinary Medical Association oath states that “veterinarians will use their knowledge and skills for the benefit of society, promote animal health and welfare and relieve suffering, protect public and environmental health and advance comparative medical knowledge, whilst improving their own knowledge and competence and upholding the standards of the profession” [5]. Similarly, the UK’s Royal College of Veterinary Surgeon’s Code of Professional Conduct for Veterinary Surgeon’s outlines veterinarian’s responsibilities to animals, clients, the profession, the veterinary team, the RCVS and the public [26].

According to the Canadian Veterinary Medical Association’s principles, veterinarians must practice “within their own area of competence” (III Professional Responsibilities, III.A.4) and “report to the appropriate authority any unprofessional conduct by colleagues” (III.C. Veterinarians’ responsibilities to the profession, 3) [6]. In Canada, where animal welfare oversight is largely enforced at a provincial level, both provincial legislation as well as provincial veterinary licensing bodies contain similar requirements for veterinary reporting [27,28].

Yet these principles may be challenged in practice and it is not always clear to the veterinarian where their responsibilities lie. For example, a veterinarian who is instructed that cases must be dealt with as far as possible “in-house” may be reluctant to refer, or one whose employer struggles with addiction that impairs his or her performance may be fearful of ramifications for reporting unprofessional conduct. Indeed, there may be little guidance as to how to act in such a case. For example, the Australian Veterinary Association’s Code of Professional Conduct (currently under review) states that “Veterinarians who become aware of misconduct, or unprofessional or discreditable conduct by a colleague should take such action as seems appropriate in the circumstances” [29].

### 3.1. Ethical Decision-Making in Veterinary Practice

One of the key areas of conflict in veterinary practice is conflict between the interests of the animal or patient and the interests of the client, who typically is paying for treatment. In most jurisdictions, the animal is legally the property of the owner. Therefore, an owner may request humane killing of an animal with a treatable condition. Should the veterinarian proceed with the request even if they disagree? Or should she take a role as patient advocate. This is what Rollin calls the “fundamental question of veterinary ethics”: “to whom does the veterinarian owe primary obligation: owner or animal?” [1]. 

The veterinarian’s actions can fall into what Rollin refers to as either the garage mechanic or the paediatrician model, based on the moral value of the animal. In the garage mechanic or human-centred (anthropocentric) model, the animal’s needs are not directly taken into consideration. Conversely, a veterinarian in the paediatrician model would primarily look after the well-being of the animal and discuss potential ethical concerns with the owner.

Many veterinarians claim to follow the paediatrician model [1,30]; however, they may fail to truly advocate for animal welfare in practice [31]. Veterinarians embedded within certain animal production industries may find it particularly challenging to separate their ethical obligations to animals from their professional responsibilities to the corporation within which they are employed.

Another question is whether ethical dilemmas are common in veterinary practice. A true ethical dilemma arises when veterinarians have competing responsibilities with no obvious way to prioritise one over the other [32]. In addition, a more controversial definition of an ethical dilemma is when there is a clear ethical choice but it is challenging to execute due to contextual factors (i.e., client interests) [32]. In reality, many ethical dilemmas are “solved” by prioritising the interests of the client over the interests of the animal. This is facilitated by legislation that reinforces the status of animals as property, without equivalent legal (and by extension, moral) standing to humans. If animals had equal legal and moral standing, it would be difficult to justify their use, for example, as sources of food or fibre. 

Veterinary decisions can also be driven by personal ethical viewpoints that may vary according to the type of client-veterinarian relationship. In human and veterinary medicine, patient-clinician relationships have generally moved from a paternalistic (clinician is responsible for all-decision making) towards a shared decision-making model. This can be challenged by the degree of involvement from the client and also the increased awareness and expectations from formally and informally (i.e., internet) educated clients [33,34]. 

Regardless, there are numerous ethical issues that arise in practice that create moral stress. Moral stress is defined as “the experience of psychological distress that results from engaging in, or failing to prevent, decisions or behaviours that transgress, or come to transgress, personally held moral or ethical beliefs” [35]. Several studies have found that killing of healthy animals, euthanasia of sick animals, dealing with clients with financial limitations and being asked to continue treatment when the veterinarian believes that euthanasia is indicated are all experienced as stressful situations by veterinarians [36,37,38].

Some of the above-mentioned issues could be the result of poor communication skills in veterinary practice. The latter is a well-recognized problem in human medicine with lesser or greater degree of professional health care consequences including malpractice claims. During stressful and complex situations, clients look for professional medical and non-medical advice, in which the veterinarian usually plays an authority figure. In addition, the increasing and rapidly changing position of companion animals as family members further accentuates the veterinarians’ responsibilities towards the animal and the client’s well-being and needs [34,39]. Chronic medical cases can be especially challenging. Clients may have been willing to have their family pet undergo weeks to months of expensive therapy or therapy with challenging side effects in the hopes of effecting a cure. They may find it difficult to accept the point at which a condition changes from being potentially curable to being definitely incurable. Sometimes veterinarians may feel responsible, in part, for the downturn in an animal’s condition, because a treatment did not work as hoped for a particular patient. As a result, they may be reluctant to make further recommendations for other treatments or euthanasia, despite having proceeded in the case using their best knowledge and skills to manage the condition. It may also be challenging to make further recommendations to a client who is lashing out because of grief over the impending loss of a beloved pet. There are many ethical parallels between these types of veterinary ethical dilemmas and parental acceptance of terminal illnesses in human paediatric patients [40]. 

Veterinarians may be concerned about speaking up about ethical medical concerns for fear of offending or alienating the client, for concerns about potential ramifications for their employment or because of conflicts of interest. In addition, lack of ethical literacy may impact their ability to articulate or justify their concerns [9]. On the other hand, veterinarians who fail to speak up risk being accused of weak morality or being complicit in animal welfare problems [33]. In recent years, curricular changes in veterinary teaching programs and continuing education offerings have been made to address some of these issues but the impacts of these interventions (for example, on the moral reasoning abilities of veterinary graduates) are yet to be seen [41]. Communications skills are usually an informal practical experience that can be difficult to assess or include in the curricula but they are the highlight of a good client-veterinarian relationship [34,42]. 

To circumvent being blamed for poor case outcome and to side-step being labelled as paternalistic, some veterinarians may avoid providing an opinion on which treatment approach they would recommend. Instead, they present a menu of treatment options to the client and then stand back to allow the client to select their preference. This “vending machine medicine” approach [38] provides the client with complete autonomy in decision-making but it also allows the veterinarian to avoid their professional responsibilities to the patient—an animal possibly with a declining state of well-being. While it may be true that the ultimate decision for care rests with the client, clients often seek guidance and support from the veterinarian when faced with difficult decisions, and, in fact, may expect veterinarians to exert their Aesculapian authority in these situations [43,44]. It has been suggested that the veterinarian must be highly attuned to the client needs and be able to balance the client’s individual preference for decision autonomy, shared decision-making and decision delegation in determining patient outcome [45]. However, in countries in which there is no legal framework for veterinarians to intervene to prevent terminal suffering of an animal, the patient’s welfare must remain a foremost consideration of the veterinarian and the veterinarian may be morally obligated to attempt to influence client decisions [44].

Inevitably, these cases can be stressful for the veterinarian as well as the client. There is little research on the efficacy of interventions in reducing moral stress; however, gathering evidence and speaking up or communicating concerns have been suggested to be beneficial in the human medical literature [21,46,47].

### 3.2. Ethics and Animal Welfare in Veterinary Practice

Animal welfare is a multidisciplinary science and researchers, veterinarians, clients, producers, consumers, politicians and others commonly use the term to describe the states that they hope to optimize in animals. Despite common usage of the term, there is no standard definition since animal welfare can be studied from different, overlapping approaches (for example, biologic function, affective state and ability to live naturally) and the definition will largely depend upon individual values and the emphasis placed on each of these three parameters [48]. When a commitment to uphold high standards of animal welfare is put into practice, the veterinary team has the potential to play an important role in animal advocacy and if there is a mature understanding of these concepts, they also have the ability and ethical imperative to be aware of and identify other human-animal interests. Rollin [1] describes animal welfare as “what we owe animals and to what extent” but also emphasizes the important role of veterinarians in ensuring good animal welfare by indicating, “… it is the natural ethical responsibility of the veterinarian to lead in putting animal welfare into practice.”

Despite similar biologic function and life interests of animals, the degree to which animal welfare is given consideration depends upon the species, purpose (e.g., production, research, companion, entertainment, etc.), local regulations, client, veterinary practitioner and individual religious and cultural values. Professional veterinary medical training provides veterinarians and veterinary technicians with tools for providing medical care, and, increasingly, include those needed for identifying and assessing animal welfare risk factors [13]. Yeates describes this role of veterinary oversight of animal welfare as a key veterinary privilege [49].

In many developed countries, there are four primary forms of guidance that veterinarians use to inform their standards of practice, including the advice and expectations related to animal welfare conveyed to clients and patients at the individual and herd or group level. These are found (i) in the legislative and regulatory requirements for a particular animal or animal-related industry in a given jurisdiction, (ii) general and specific veterinary professional standards and guidance documents, (iii) generally accepted animal husbandry practices and (iv) livestock industry animal welfare assurance programs. These will be each be discussed in more detail in the following paragraphs.

The legal requirements for good stewardship of animals, such as those promulgated by the OIE, European Food Safety Authority (EFSA), U.S. Department of Agriculture (USDA), etc., are a reasonable starting point for ensuring good animal welfare, as these regulations are generally reflective of societal expectations for basic animal care. In many countries, federal and state or provincial animal health, welfare and transport acts and regulations dictate the minimum expectations for animal care. For food animal species, many of these regulations extend beyond the farm gate. As such, food and production animal veterinarians need a full understanding of all aspects of the production, transportation and marketing systems in which their clients and patients are found to be able to provide comprehensive advice and education to ensure good animal welfare during all phases of an animal’s life. 

Beyond federal and state/provincial laws and regulations, professional standards for veterinarians are set through national, provincial and local veterinary associations and statutory licensing bodies. Most licensing bodies in developed countries require veterinarians to actively participate in continuing education (also known as continuing professional development or CPD) as a condition of licensure (e.g., see the RCVS policy on continuing professional development, https://www.rcvs.org.uk/lifelong-learning/continuing-professional-development-cpd/). Animal welfare is increasingly present as a theme in veterinary conferences, journals and discussion fora. Professional standards and expectations with respect to animal welfare reflect a peer-reviewed and evidence-based approach to considerations of aspects of veterinary medicine including animal welfare and ethical decision-making (for example, the American Veterinary Medical Association Guidelines on Euthanasia) [50]. 

In terms of nationally accepted animal husbandry practices, there may be published national standards, such as those produced by the Canadian Veterinary Medical Association (CVMA), i.e., the CVMA Kennel Code and Cattery Code, which cover general husbandry expectations for these companion animals [51,52]. These documents are used to establish standards for husbandry and care of dogs and cats in boarding kennels, shelters, pet stores and other places. Plans are underway at the CVMA to develop a code of care for small mammal species, such as rabbits and ferrets. For large animal species, the National Farm Animal Care Council (NFACC) in Canada has published Codes of Practice for the care and handling of 15 different food or fur-bearing animal commodity groups, as well as a general transportation standard [53]. These codes have been developed in consultation and cooperation with livestock industry stakeholder groups, including scientific advisory committees, farmers and food industry representatives, animal welfare advocacy groups and government representatives. Such codes provide an agreed upon set of standards for animal husbandry and while adherence to them is voluntary, the codes have been used to enforce charges for animal welfare violations in some provinces. 

Assessment drives change and ensures adherence to accepted practices of care, thus more mature animal welfare oversight programs include an assessment tool to permit benchmarking of progress. In Canada, NFACC has also developed and published a framework for animal care assessment from which livestock commodity groups can develop animal care assessment tools and programs. Many of the Canadian livestock commodity groups have adopted the species-specific codes as standards and have begun the process of adoption and implementation of animal care assessment programs. Many USA and Canadian livestock commodity groups also have self-mandated quality assurance programs that include animal welfare components, such as the Dairy Farmers of Canada Canadian Quality Milk Pro-Action Program and the Pork Quality Assurance Plus program in the U.S.A. [54,55]. Veterinary awareness and participation in these programs is mandated and the programs require training and ongoing communication between veterinarians and their clients to ensure consultations with producers. The goal of most of these programs is to establish, then meet and exceed animal care and welfare program benchmarks, by using both on-farm animal-based measures and evaluation of standard operating procedures and records related to appropriate mature and immature animal care and comfort, health and behavioural monitoring, culling procedures and on-farm euthanasia. Industry-led assessment and assurance schemes are not currently available for companion animal care and many other animal-related businesses (e.g., rodent breeding operations for reptile feeding, aviary breeders for the pet trade, etc.) as well as various animal holding facilities (e.g., exotic animal sanctuaries and private roadside zoos), except through specific veterinary clinic accreditation schemes (e.g., the American Animal Hospital Association (AAHA) [56]. For these species, there may be more of a reliance on general quality of life and physiologic measures for welfare assessment, as well as a post factum review of forensic evidence in more egregious animal abuse and neglect cases [57].

Where they exist, animal regulatory frameworks, a clear understanding of acceptable animal husbandry practices, nationally accepted peer-reviewed standards of animal care and animal welfare assessment schemes all provide a strong sense of acceptable animal care and veterinary practice standards for veterinarians and their clients; however, these are only theoretical frameworks. To truly and positively impact animal welfare in the long run and meet societal expectations for the profession, the veterinarian must combine this knowledge with their ethical obligations to animals and actively speak up and ask questions when they observe or suspect that animal well-being is compromised.

## 4. Failure to Meet Animal Welfare Needs

Developing useful welfare guidance documents and schemes and then actively engaging clients in discussion, education and implementation of these legal and voluntary standards and available animal welfare assurance programs are ongoing, evolving and necessary processes. For reasons mentioned previously, despite exposure to these various tools during veterinary medical training, veterinarians may not always act on their knowledge and in the best interests of animals in their care. When interviewed about processes underlying medical decision-making, veterinarians frequently admitted to making decisions about their clients that were unspoken and that could adversely impact animal welfare [31]. This included classifying clients as ‘good’ or ‘bad’ in terms of willingness to pursue treatments or make payments, as well as holding perceptions about whether clients could afford certain therapies or techniques for their animals, such as analgesia. Thus, if the veterinarian judged that the client could not afford analgesia or would not wish to pay for it, analgesia was not offered, thereby depriving the client the opportunity to elect analgesia use [58]. Limiting disclosure of options because of client categorization can both limit client care of their animals and the veterinarian’s ability to promote animal welfare [40]. It can also be profoundly uncomfortable to ask difficult and sensitive questions related to possible neglect of an animal, particularly of longstanding and trusted clients and may result in ‘emotional blocks’ or cognitive dissonance within the veterinarian [58].

In a large veterinary referral or teaching hospital with both small and large animal capabilities or at a multispecies livestock auction, it is common to observe on a near daily basis instances of animals suffering from poor welfare and for which poor decisions have been made by clients, their veterinarians or both. The true prevalence of these types of issues is unknown, since there is rarely open discussion about welfare failures within the veterinary literature and these would need to be considered against a denominator of all client cases treated. Almost all of these cases are preventable, although the etiology of each problem may be slightly different. Note that there is no single and correct framework for approaching such cases, as each must consider the needs of the animal and the context in which the case is presented. The following represent examples of different root causes resulting in poor animal welfare. These cases were selected for their complexity and apparent challenging translation of veterinary ethics theory into practice. These cases are not representative of all types of ethical challenging situations in veterinary medicine but we have tried to cover a range of possible areas of ethical conflict, such as those seen in companion and food animal practice and exotic animal sanctuary medicine. 

### 4.1. Example 1: Poor Food Animal Transportation Decision

In this case, a severely lame cull cow, as depicted in Figure 1, was transported from a farm to a livestock sales barn. In most developed countries, severely lame cattle are considered to be unfit for transport, save for the purpose of seeking veterinary medical treatment (for example, Livestock Transport Requirements [59]). This is because it is accepted that transportation of severely lame cattle through normal production channels (i.e., to livestock auctions or slaughter facilities) has reasonable potential to cause undue pain and suffering, further compromising individual animal welfare.

Livestock sales and their associated regulations are observed and enforced, respectively, by regulatory veterinarians with responsibilities for segregating, inspecting and making decisions about animals that arrive and are deemed to be compromised. In this case, the cow was unloaded at the sales barn and found to be severely lame and non-weight bearing on her right hind leg. The cow was segregated by sales barn staff for veterinary inspection. The regulatory veterinarian determined that the cow was dull, extremely thin, reluctant to move, unable to keep up with a group of conspecifics, moderately lame in her left hind and right front limbs with severe swelling of her right carpus and non-weight bearing in her right hind limb while standing. The cow was euthanized and submitted for post-mortem examination to investigate the cause of the lameness and to offer insight into the chronicity of any causative lesions. Post-mortem examination of this cow revealed severe, chronic bilateral digital dermatitis and marked heel horn erosion in the hind feet. Of note, digital dermatitis is part of a spectrum of, mostly treatable, hoof lesions in cattle [60].

This case represents an example of how on-farm decisions can have serious consequences for animal welfare well beyond the farm-gate. On-farm preventive medicine, lameness identification and treatment protocols, as well as culling decision procedures, all developed in consultation with the herd veterinarian and properly implemented, could have prevented this cow from developing severe lameness and poor body condition and being transported off the farm.

Veterinary involvement in the enforcement of federal and provincial regulations worked to limit further suffering at the sales barn. While the exact reasons underlying the decision to transport this cow are unknown, this example highlights a need for the herd veterinarian to provide more support on-farm. Animal transportation is a complex issue because of possible conflict between the veterinarian’s recommendations and producer’s interests as well as the availability of suitable methods for on-farm euthanasia and the cost of animal disposal [61]. Despite this, veterinarians are an integral member of the livestock production, health and welfare team and they should be strongly grounded in the ethics of care (Table 1) and they can provide resources to clients to assist them with better animal management practices (Figure 2). The outcome in this particular case demonstrates how the veterinary-client relationship failed. Improved communication and veterinary involvement in on-farm ethical decision-making could have significantly reduced the impact of disease on animal welfare.

### 4.2. Example 2: Animal Neglect

A nine-year-old Amazon parrot was presented to a referral hospital for a second opinion regarding chronic non-union fractures incurred several months previously. Following an unknown injury, the fractures were treated by another veterinarian using body bandages. When collecting the history, the client indicated that the bird used to fly in the house and they were concerned that the bird could no longer fly or ambulate properly since the injury. Radiographs performed by the tertiary care veterinarian identified several non-union fractures in the pelvis and limbs, as well as evidence of serious metabolic bone disease in the femurs, tibiotarsi and synsacrum with major skeletal abnormalities that likely contributed to the inability to ambulate. The bird was euthanized and the presence of multiple healed and unhealed fractures and other chronic bone deformities that likely arose secondary to metabolic bone disease were confirmed at post-mortem (Figure 3). The report noted that similar injuries in other birds are commonly associated with marked pain.

Metabolic bone disease is common and treatable condition in birds receiving inappropriate diets and, if untreated, can lead to bone deformities and fractures, as in this case [63]. Based on information received from the referring and tertiary care veterinarians there was a sense that the client was unable to pay for expensive treatments but there was also no indication that either had questioned the client in-depth about the bird’s diet or about how the multiple fractures had been incurred. Veterinary and medical care can often be influenced by tunnel vision or decision-making biases, in which the client and veterinarian focus on diagnosing and treating a specific injury and lose sight of the overall prognosis or presence of intercurrent disease [64]. Veterinarians rely heavily on the history provided by the client as well as on physical examination findings and these must always be placed in context with their knowledge of common conditions of any given species. Regardless of the bond or level of commitment that a veterinarian may feel that a given client has towards an animal, they should not assume that the client is aware of how best to meet an animal’s needs, even in the face of longstanding ownership. There is an increasing awareness of animal welfare issues associated with exotic companion animals, largely related to the client’s lack of knowledge about appropriate care, husbandry and needs of these animals [64,65,66]. From a virtue ethics framework (Table 1), the referring veterinarian could have asked specific questions about the diet and husbandry of the bird and provided information about appropriate nutritional and husbandry needs, in addition to instituting appropriate treatment and follow-up for the fractures. By not speaking up, the veterinarian contributed to ongoing negligence and this potentially treatable condition went unrecognized for many years, culminating in significant animal suffering. Potential and disturbing questions about animal neglect and abuse were also not investigated by either veterinarian for reasons unknown. 

### 4.3. Example 3: Economic Decisions Impacting Animal Welfare

An aged female rat was presented to a newly hired veterinarian in an exotic animal clinic for a perineal mass that the client indicated had appeared only one week prior. Due to financial constraints, it was reported that the owners elected to pursue surgical debulking of the mass without further diagnostics. The owners brought the rat back to the veterinarian one month after the initial surgery, as the mass had regrown and had ulcerated (Figure 4). On physical examination, the rat was in poor body condition and had a large irregular perineal mass. Diagnostic testing was declined and the rat was euthanized. Subsequent post-mortem and microscopic follow-up indicated that the mass consisted of a rapidly growing and invasive vaginal leiomyosarcoma.

In this case, by trying to provide an inexpensive solution to satisfy the client and not taking into full consideration the circumstances of the animal, the veterinarian inadvertently created a more significant welfare problem and unnecessarily prolonged the suffering of this rat. While resulting in a short-term, cost effective and aesthetically acceptable solution, superficial debulking of a tumour without knowing the type of tumour that was present and whether adequate surgical margins were achieved was unnecessarily risky. When electing any type of empirical treatment, the client must be informed of all potential outcomes. In this case, the potential for a poor outcome due to the presence of a rapidly growing mass in a geriatric animal and in a location with significant potential for impairing normal bodily functions together with the client’s significant financial constraints should have been used to make a recommendation for euthanasia based on a utilitarian framework and the potential for animal suffering as a result of surgery and potential for tumour recurrence. The underlying reasoning of the veterinarian in the case is unknown. It could also be argued that the veterinarian was in a conflict of interest situation in that they were in a position to gain more financially from performing the surgery or that they were interested in increasing their service caseload, clinical skills or clinical reputation. Veterinarians must be prepared to speak up and provide advice to clients that is in the best interests of the animal regardless of whether they would prefer to attempt a different treatment approach. In addition, the veterinarian could have consulted with a veterinary colleague as to the most appropriate course of action and should have disclosed the necessity for pursuing further diagnostics to the client [43,67]. When clients are financially constrained (or appear to be so), as in this case, the consequences of proceeding with further treatment without the benefit of appropriate diagnostic insight must be carefully weighed against the potential for the animal to incur more significant harms. This should be a common means of influencing the decision-making process for clients and veterinarians when dealing with cancer patients [68]. Suboptimal evaluation of animals with cancer, as in this case, can be problematic. 

### 4.4. Example 4: Inability of the Client to Accept a Poor Quality of Life

A 14-year-old spayed female American Bulldog was under the care of a veterinarian at a primary clinic for chronic osteoarthritis and a large ulcerated and necrotic cutaneous mass on the shoulder of many months duration. For reasons not specified in the history, the client and the veterinarian did not pursue biopsy or staging, or analgesia and instead, the client elected to cover the thorax with a t-shirt. The dog was referred to a tertiary care centre because the mass continued to ‘split open’ and ultimately, the dog was euthanized due to poor quality of life concerns. The dog was submitted for post-mortem examination and was noted to be wearing a blue t-shirt that was heavily stained with malodorous fluid (Figure 5). When the t-shirt was removed, a large and markedly necrotic mass was present within the skin and subcutaneous tissues of the left shoulder. Histologic examination identified this tumour as a grade II soft tissue sarcoma, with no evidence of metastatic disease. Degenerative joint disease was noted in the right coxofemoral joint and several other previously undiagnosed and unrelated tumours were detected in other organ systems.

While the mass and associated wound in this case were not lethal in nature, the chronic festering quality would have contributed to long term discomfort for the animal and an inability to lie on the affected side, increased susceptibility to infection and even the ability to take this animal into public places and possibly spend time with the animal, given the extremely malodorous and oozing nature of the wound. Respecting patient or client autonomy in decision-making is a common ethical challenge for human and veterinary practitioners [40,45]. Obtaining informed consent in veterinary medicine provides legal protection to practitioners but it should not be used as a sole means for justifying their actions or inactions [67]. While being respectful of client wishes, veterinarians must also play a more active role when animal welfare concerns are present by providing tools to assist with decision-making, such as formal or informal quality of life assessments [68,69,70], to help clients to better understand and appreciate welfare problems and to avoid unnecessary pain and suffering, which takes into consideration a more utilitarian ethical framework (Table 1) Certainly, the results of these assessments can be subjective and may still require the veterinarian to speak up and advocate for the animal if further treatment cannot be pursued. 

### 4.5. Example 5: Refusal to Accept End of Life of an Animal

A 6-year-old male ringtail lemur from an exotic animal sanctuary was attacked by a Japanese macaque following a public health quarantine that was placed in the sanctuary for unrelated reasons. The two nonhuman primates had to be placed together within a relatively confined area, because there was insufficient caging to keep them separated. The sanctuary was approved by a regional SPCA office as an official site for rehoming exotic animals seized by inspectors. The injuries to the lemur were severe and extensive and included marked, multifocal bites, skin loss and lacerations, tail mutilation, bilateral hind limb lacerations and fractures and bilateral forelimb fractures. Following this emergency, the client sought medical attention for the lemur from a local companion animal practice with which they had a longstanding relationship but which did not have specific exotic animal or primate medicine expertise. Several successive surgeries were performed on this animal over the ensuing 3 weeks resulting in amputations of the tail and both hind limbs, plating of fractures in both forearms and later amputation of the right forelimb. The lemur was left in a completely non-ambulatory condition with only the pinned left forelimb remaining. The owners refused the initial veterinary recommendation of euthanasia and elected to carry the lemur using an infant sling while providing oral antimicrobials and once daily dosing of a nonsteroidal anti-inflammatory drug. Four weeks following the injury the lemur died spontaneously after a 1-day history of coughing and diarrhea and was submitted for post-mortem examination. At post-mortem, an open comminuted fracture was noted in the left forearm as well as evidence of chronic ulceration on the dorsum of the distal trunk, suppuration around surgical wounds, acute enteritis, aspiration pneumonia and early renal failure (Figure 6).

Because of the severity of the injuries, consideration should have been given to immediate euthanasia of the lemur at the time of initial examination, according to utilitarian and ethics of care viewpoints (Table 1). Nonhuman primates can suffer myoglobinuric nephropathy secondary to severe trauma from fighting [71]. In this case, the client was angry about the unnecessary public health quarantine that had resulted in trauma to this animal and the client was also unduly influenced by their deep attachment to and affection for the lemur. The veterinarian subsequently admitted that the clinic had discussed the ethical and quality of life implications of successive amputations for the lemur but indicated that ultimately, the clinic owner had made the decision to proceed with surgeries when the client refused euthanasia, particularly since there were no financial limitations for the client. The veterinarian also admitted to insufficient knowledge about primate medicine and care as well as feelings of unease with how this case had been handled. When subsequent problems developed in this animal over the course of the three weeks, the clinic felt heavily invested in the case and continued to provide medical and surgical support, up to three days before the lemur was found dead. Subsequently, the SPCA mandated that better housing and husbandry conditions be instituted for the remaining animals at the sanctuary. 

This case represents a number of ethical challenges that can occur in veterinary practice. Junior veterinary associates in a clinical practice may not be comfortable with the ethics of certain medical decisions made by colleagues but may feel unable to speak up because of respect for the opinion of more senior veterinarians, because of fear of reprisal and loss of income if they express a minority opinion and are subsequently fired and because of insufficient confidence with their own knowledge and skill set to know whether proceeding in a certain way with a case can or should be done. Pride and overconfidence can also adversely impact patient welfare and result in permanent harm or death [72]. Knowledge of orthopedic surgery in companion animals does not necessarily translate to adequate knowledge of orthopedic surgery in less familiar species, such as nonhuman primates. In this case, the veterinarians elected not to pursue reasoning with the aggrieved client and profited from the situation, resulting in a very poor welfare outcome for the lemur. Using a virtue ethics framework (Table 1), the veterinarians would have been able to articulate the limitations in their own skillset and knowledge, as well as the potential conflict of interest for significant financial gain and sought further outside advice and support before proceeding with the surgeries. Peer interactions and clear communications of possible outcomes, welfare concerns and an optimal course of action are important to discuss with the client. As a last resort, veterinarians can also invoke legal reporting requirements if there are significant client conflicts that create major animal welfare risks.

### 4.6. Example 6: Failure to Provide Timely Veterinary Care Follow-Up

A dairy farm with 60 milking Holsteins experienced a serious and sudden barn fire. All cows, including milking and dry cows and pregnant replacement heifers were removed from the burning structure, although this occurred under extreme circumstances. The barn was completely destroyed in the fire and a press release issued afterwards indicated that no human or animal lives had been lost in the blaze. The herd veterinarian was called to the farm and provided fluids and several doses of nonsteroidal anti-inflammatory drugs to a pregnant heifer that had been injured in the blaze, although because of the detritus clinging to the heifer’s hair coat, the true extent of the injuries may not have been immediately obvious. The milking cows were moved to a neighbouring farm for the next month and the remaining dry cows and replacement heifers were moved to a distant pasture on the property, with minimal daily attention, while the devastated farmer and family focused on rebuilding the barn. Almost four weeks after the event, the herd veterinarian revisited the farm and found the previously treated heifer with severe burns on the dorsum. The cow was administered a nonsteroidal anti-inflammatory drug and was shipped to a nearby veterinary referral centre. Upon arrival, the receiving veterinarian ordered immediate euthanasia of the heifer on the truck because of significant concerns about poor welfare and unrelieved pain and distress. At post-mortem, the heifer was noted to have extensive third degree burns and eschar covering approximately 60% of the dorsum (Figure 7).

In this case, the veterinarian admitted to forgetting about the cows in the back pasture, in part, because they were not convenient to access and because of the overall busyness of their practice. The client had also largely ignored these animals in the aftermath of the fire, because of their shock surrounding the incident, severe economic straits until insurance money was available and their ongoing attention to rebuilding their barn and several other outbuildings. The lack of attention and follow-up in this case led to severe suffering of the affected animal. The veterinarian subsequently referred the animal as a teaching donation to minimize any costs to the client associated with deadstock removal as well as the difficulty that would have arisen to remove the carcass from the remote pasture. This was a very difficult case because of the complexity of circumstances. From a utilitarian point of view (Table 1), the heifer should have been euthanized at the original visit, because of the extent of the injuries suffered and because it was known at that time that medical care of the animal could not be managed intensively. Conversely, from a consequentialist approach, the heifer was originally kept because she was carrying a genetically valuable foetus and if she had survived, the end may have justified the means. For this approach to be justifiable, the heifer would have needed to be monitored carefully and treated intensively. Poor decisions for animal welfare were made in this case because of the veterinarian’s sympathy for and assumptions about client circumstances and because of inattention to their ethical duty to the animal and an inability to speak up and advocate for the heifer’s welfare. It is unknown whether the veterinarian delayed conducting a revisit sooner because of concerns about the client’s ability to pay for the call. The receiving veterinarian immediately recognized the severe and adverse welfare state of the heifer and acted promptly to relieve further animal suffering.

## 5. Applying Ethics to Veterinary Practice

Veterinary decision-making will always be complex and messy and the ethical reasoning underpinning a course of action cannot be simply addressed by turning to an algorithm or flowchart, as often one or more courses of action may be reasonable for a given situation. Additionally, the fact that there is no clear metric to aim for may make the concept of ‘ethical veterinary practice’ seem like a nebulous goal. The value of emphasizing ethics in everyday clinical practice is that it helps the clinician to reflect on their course of action, it empowers clinicians to advocate for their patients and it is critical for informing policy—policy for the profession and for animal welfare [73]. Ultimately, veterinarians have an ethical obligation to provide good care for their patients and clinics and universities must provide initial training in ethical decision-making and then nurture a culture that enables veterinary students and practitioners to speak up. Without this, there is a gap between the theory of ethical veterinary practice and its actual application, which, as has been discussed, can lead to significant moral conflict and burn-out as well as significant animal welfare issues, as per the examples provided. Kong describes the creation of an ethics community in human medicine to nurture ethical reasoning and moral imagination [73]. An ethics community is created when academics and clinicians are sensitive to ethical issues and encourage ongoing dialogue in a safe environment in which practitioners can speak openly about their clinical ethical concerns [73]. In veterinary medicine, this could include nonjudgmental and peer-to-peer discussions with like-minded colleagues, as well as veterinary ethicists and academic researchers. 

Moore describes a “common-morality theory” in bioethics characterized by pre-theoretical common-sense ethical judgment that acts as starting point of view for most ethical frameworks [74]. This background in ethics knowledge and knowledge of veterinary ethics frameworks and approaches during veterinary medical education will nurture ethical reasoning for future practitioners. For those already practicing veterinary medicine, continuing education focusing on medical skills and ethical decision-making as well as peer-to-peer discussions can facilitate higher levels of moral reasoning in complex cases and in determining the most appropriate course of action [75]. In the end, ethical medical reasoning is a highly reflective process characterized by a continuous assessment of the advantages and disadvantages of each framework over time [73,74].

## 6. Conclusions

Ethical and moral issues arise commonly in all spheres of veterinary practice. Ethics education during veterinary training may help to improve ethical sensitivity and equip veterinarians with frameworks and approaches that support ethical decision-making. These also provide clear justifications for “speaking up” in ethically challenging situations. Good communication skills lie at the heart of the veterinary-client relationship and veterinarians must not assume a passive role when serious welfare matters are at hand. Advocating for animal welfare may not be comfortable and may, at times, require courage but is necessary to advance the veterinary medical profession and to improve human regard for animals as sentient beings. Further research and discourse on veterinary ethical issues may improve veterinary teaching and translation of ethics theory and reasoning into applied practice.

## Figures and Tables

**Figure 1 animals-08-00015-f001:**
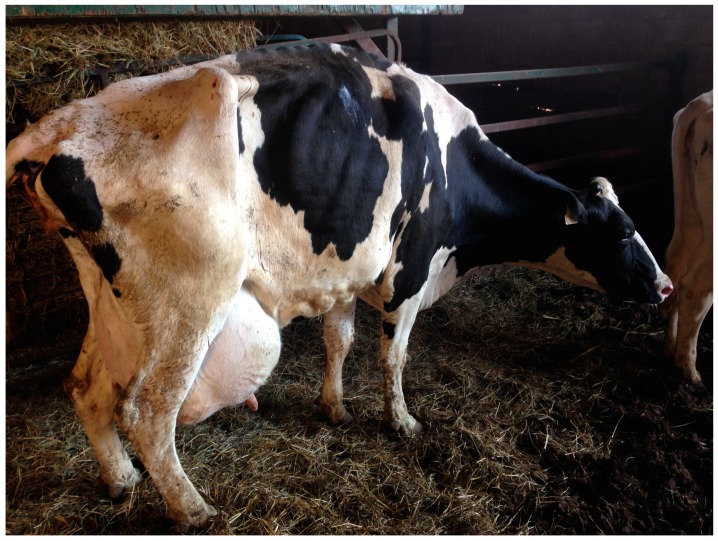
Severely lame dairy cow in very poor body condition presented at a livestock sales barn.

**Figure 2 animals-08-00015-f002:**
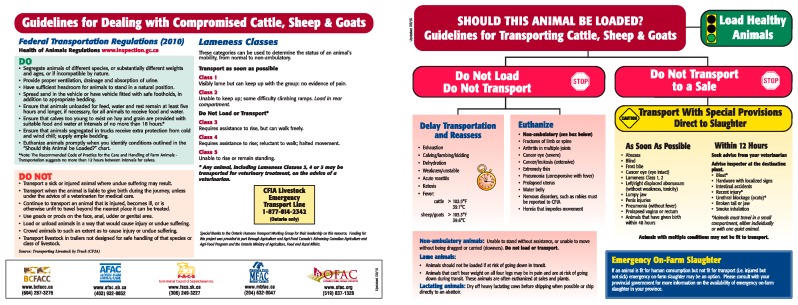
Example of guidelines available to assist producers and veterinarians with decisions surrounding animal transportation [62].

**Figure 3 animals-08-00015-f003:**
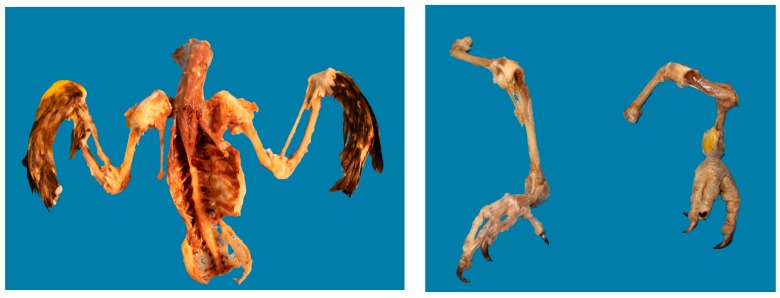
Chronic clockwise deviation of the pelvis and sacrum and multiple chronic rib fractures (**left**); Chronic non-union mid-diaphysis fracture of the left tibiotarsus with firm callus and chronic non-union fracture of the proximal metaphysis of the right femur and tibiotarsus (**right**).

**Figure 4 animals-08-00015-f004:**
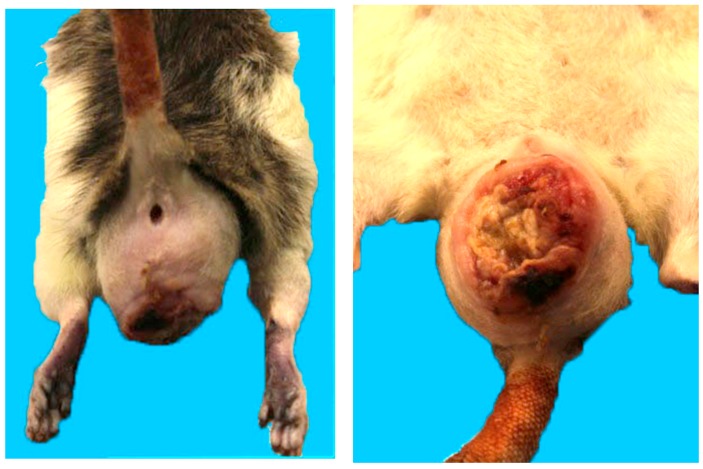
Dorsal view of a perineal mass in a mature female pet rat (**left**); Ventral view of ulcerated perineal mass in the same rat (**right**).

**Figure 5 animals-08-00015-f005:**
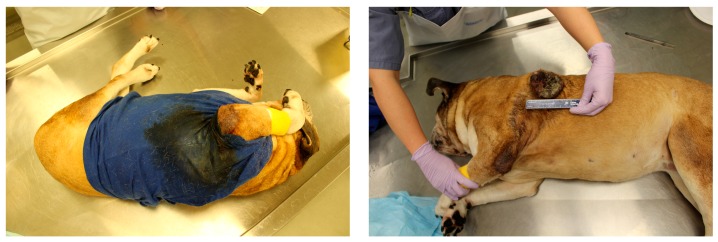
Lateral view of the dog with the t-shirt on demonstrating staining from the underlying necrotic mass (**left**). T-shirt removed and the large ulcerated mass revealed (**right**).

**Figure 6 animals-08-00015-f006:**
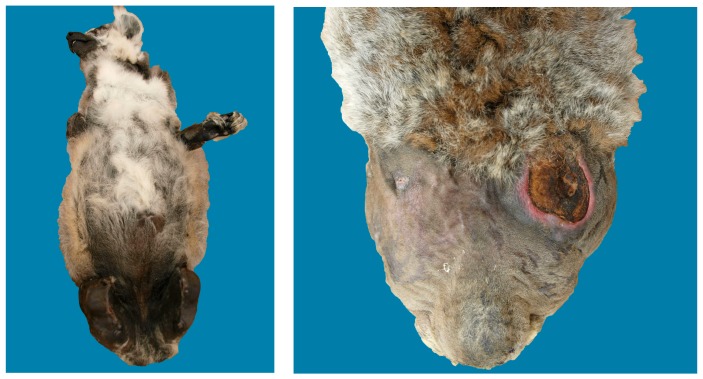
One-armed ring-tailed lemur presented for post-mortem examination (**left**); A deep suppurating ulcer was present on the dorsal trunk (**right**).

**Figure 7 animals-08-00015-f007:**
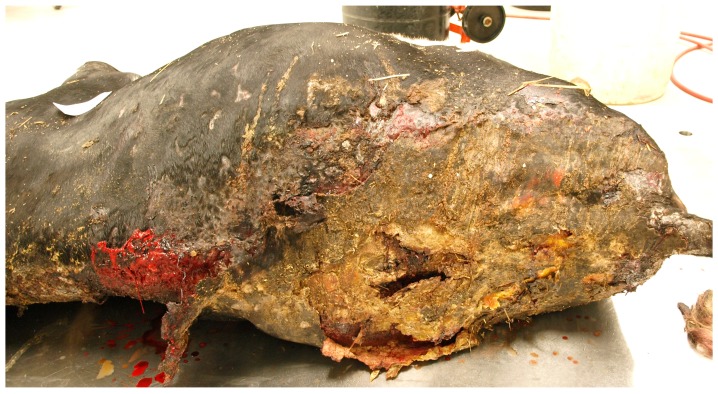
Holstein heifer with third degree burns to dorsum one month following a barn fire.

**Table 1 animals-08-00015-t001:** Strengths and limitations of key ethical frameworks and approaches taught to veterinary students and an interpretation of how “speaking up” may be conceived according to each framework or approach. Adapted from Mullan and Fawcett [21].

Ethical Framework/Approach	Explanation	Strengths	Limitations	Speaking up
Utilitarianism	Ethical decision making should aim for the greatest good (maximal pleasure, minimal suffering) for the greatest number of stakeholders. This is a form of cost: benefit analysis.	Stakeholders include any being with a capacity to suffer.Impartiality (in theory): any stakeholder is morally equal to any others.The consequences of decisions/actions are taken into account—rule-bending or breaking is allowed if it leads to a good outcome.	Can be used to justify exploitation of minorities, such that they bear the costs while the majority enjoys the benefits.Focus on maximisation of benefits does not address fairness of distribution of benefits.Does not recognise the rights of individual stakeholders.Can be used to justify immoral means to an end.Can be difficult weighing costs against benefits.	Whether a veterinarian decides to speak up depends on the consequences. This should lead to a better outcome (less suffering, more pleasure) for the greatest number of stakeholders.
Deontology	Ethical decisions are correct if they conform to a moral rule or norm.	Takes the intentions of the decision maker/s into account.Recognises the rights of individuals (though not necessarily animals)Consistent with language of legislation, professional codes of conduct	Does not take consequences of a decision/action into account.Inflexible—one cannot tell a “white lie” to achieve a good outcome.Offers no guidance when it comes to managing conflicting rights.Tend to be phrased as negative constraints on our actions.Promotes “loophole” seeking.	A veterinarian has a duty to speak up, regardless of consequences. This may coincide with Professional Codes of Conduct and legislation.
Contractarianism	Ethical rules, norms and obligations derive from an explicit or implied contract or mutual agreement.	Acting ethically is in one’s self-interest.	Limited by who can enter into a contract.Favours human interest and at best confers indirect rights to animals.	A veterinarian must speak up (or not) if they are contractually obliged or have agreed to do so (or not do so).
Virtue ethics	Sound ethical decisions flow from having a virtuous character.Virtues are character traits that are reliably present in individuals. Examples include compassion, discernment, trustworthiness, integrity, conscientiousness [23], initiative, self-discipline, responsibility, integrity and accountability [24].	Recognises that emotions are key in ethical sensitivity and decision making.Intent or motivation of the decision maker is what matters.Recognises moral strength, i.e., the virtuous veterinarian does not blindly follow the rules.Flexible—virtuous people may behave in different ways despite similar circumstances.Corresponds with societal expectations about professionals.Emphasis on personal development and reflection.	May not act as a stand-alone framework.There can be conflict between virtues (for example, loyalty and honesty).Virtues may manifest very differently according to the role of each stakeholder. For example, when deciding whether to treat an animal, a veterinarian decides according to what a virtuous vet would do—while the owner decides according to what a virtuous owner would do. There is scope for disagreement.	A virtuous vet may speak out because they are trustworthy, however the virtuous vet is also discerning and may choose not to speak out in certain circumstances (for example those that may cause suffering).
Ethics of care	Unlike other “impartial” theories, recognises our relationships and obligations of care that follow from these.	Reflects the fact that animals and other beings are dependent on us.Provides for morally defensible protection of and distribution of key resources to our loves ones (including animals)Demands that animals in relationships with humans are treated in caring ways.Capitalises on existing moral sentiments about people and animals, particularly family members and companion animals and formalises a duty of care that many already recognise.	Not a stand-alone ethical framework.No consensus about what it means to “care” for others.There remain moral limits to the care we may legitimately expect from others.	A veterinarian must speak up for the interests of those with whom one has a relationship (for example, clients and patients).

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
