# Peer review of "Speaking Up: Veterinary Ethical Responsibilities and Animal Welfare Issues in Everyday Practice"

_animals, 2018, doi:10.3390/ani8010015_

Round 1

Reviewer 1 Report

Brief summary - The submitted paper provides a retrospective outlook on the teaching of veterinary ethics (why and how) and applies ethical concepts to address animal welfare case scenarios. Bearing in mind the dearth of available literature on these subjects, this is arguably a valuable contribution to the field of veterinary ethics.

Broad Comments – Despite the relevance of the submitted paper, its aims are not clearly defined; in my view, the submitted paper aims at providing a bird’s eye view on veterinary ethics education across the globe, which will then inform the resolution of case scenarios in animal welfare ethics. Unless considered otherwise, these features should be clearly identified in the abstract, the introduction and the conclusion. Similarly, the structure of the paper could be improved; the discourse is often fragmented and contents could be arranged differently. The case scenarios need to be revised. The reference list is broad but some key references may be missing (cf. specific comments).   

Specific Comments

Title - the title is concise but arguably not entirely clear. The title should emphasise the scope of the paper (i.e. the role of ethics in informing veterinary responsibilities towards animal welfare).

Simple summary – This section reflects poorly the content of the paper and should be revised.

2. Ethics in Veterinary Medicine – This chapter is about veterinary ethics education. Shouldn’t the word ‘education/teaching/training’ be part of the title?

 2.1 Why is ethics taught? This section requires restructuring and further insight. The first paragraph (lines 43-46) is not clearly connected with the following paragraphs and something is missing in between. Why do the authors think that veterinary ethics teaching in the 1970s-80s was about professional etiquette and not about euthanasia? What has changed since? The section then examines recent advances in veterinary ethics education in Europe, Canada and the US. Would you consider including Australasia for a truly global perspective? This geographic sequence could be repeated in the next section. Furthermore, research from Main et al (lines 75-79), is about how ethics is taught and should be part of the next section. Finally, better use of reference 8 - one of the few attempts to explore why veterinary ethics is taught - is advised.

2.2 How is veterinary ethics taught?

Line 92 -… (including ethical frameworks and approaches), laws….

Line 93 – Ref [8]. This reference seems wrong. The authors probably meant https://doi.org/10.1136/vr.102553. Furthermore, this reference could be used to guide the remaining of the section and help justify the choice of the case scenarios (cf. below).

Line 116 - --- such as In Practice’s Everyday Ethics …

Lines 132-143 – Again, could codes of conduct from jurisdictions other than Canada be used to inform this section?

Line 149 onwards – I think that by now the reader has been guided towards a next step of ethical enquiry and that a new section on something like ‘veterinary ethical decision-making’/’ethical responsibilities’ should start here. Paragraph in lines 156-162 should be moved further down.  

Line 345 onwards – The authors chose 7(6?) examples of prominent veterinary ethical issues with animal welfare implications. Several issues arise: What guided the choice of these case scenarios? Are they supposed to be genuine ethical dilemmas or just challenges? Why focusing in cases of animal welfare and not in cases of professionalism (which are also part of vet ethics)? Why choosing two exotic and one wild species and not presenting horses, pigs or poultry? Moreover, further efforts should be made to connect the case scenarios with the previous sections on vet ethics education (in terms of e.g. ethical frameworks, professional guidelines, regulations…). For example, virtue ethics, ethics of care and codes of conduct have not informed any of the cases. You could probably apply different frameworks to different cases as to illustrate their rationale. Alternatively, some of the information provided in the previous sections (especially in 4. Ethics and animal welfare in veterinary practice) could be fed into the case scenarios.

Example 1 – I think that this example lacks insight. A recent investigation from Ireland unravelled the complexity of issues involved when deciding to transport and slaughter injured livestock https://doi.org/10.1186/s13620-017-0102-0, including the role of client pressure on clinical decision-making, the availability of on-farm emergency slaughter and defective regulations.

Example 2 – Line 412 – References should be added pointing toward the incompetence of exotic pet owners (e.g. https://doi.org/10.1016/j.jveb.2017.07.005).

Example 3 – The authors should allude do the concept of that ‘X-raying pockets’ (i.e., trying to predict how much the client will want to pay) may prevent the client from having a full awareness of the available treatment options (c.f. https://doi.org/10.1136/vr.158.2.62 and Ref. 31)

Example 4 – The concept of autonomy should be discussed in light of the concept of informed consent (c.f. https://doi.org/10.1007/s41055-017-0016-2). What should have been done differently?

Example 6 - ???

Example 7 – This is a strange scenario and not entirely clear. Did the vet overlook the extent of the burns in his first visit? When did he forget the remaining cows? I think this is more a case of professional negligence than truly an ethical challenge. Couldn’t this case be replaced by an alternative case on broiler welfare? (e.g. vet detected lame broilers, prescribed new feed, one week later the broilers couldn’t stand up; decided for euthanasia, despite substantial economic loss…).

Conclusion – In its present form the conclusion is insufficient. I suggest that the information provided in the Simple Summary should be moved to the conclusion. I am not totally in agreement to separate between ‘animal welfare issues’ and ‘ethical dilemmas’ (again, are the case scenarios one, the other, or both?). Please elaborate on the statement “all veterinarians require a strong ethical framework”; it seems to imply that we just need to pick one from Table 1 and we’re done! The authors should point toward the future, including recommendations in terms of educational needs in animal welfare science, ethics and law for veterinary students /veterinarians. Regarding the case scenarios, the authors should clarify that there are arguably several possible approaches to successfully resolve them and that the presented suggestions are not the only solutions.

Author Response

Reviewer 2

Comments and Suggestions for Authors

Brief summary - The submitted paper provides a retrospective outlook on the teaching of veterinary ethics (why and how) and applies ethical concepts to address animal welfare case scenarios. Bearing in mind the dearth of available literature on these subjects, this is arguably a valuable contribution to the field of veterinary ethics.

Thank-you for your comments and insight; we hope that with the suggested edits and comments we have provided an original and valuable contribution.

Broad Comments – Despite the relevance of the submitted paper, its aims are not clearly defined; in my view, the submitted paper aims at providing a bird’s eye view on veterinary ethics education across the globe, which will then inform the resolution of case scenarios in animal welfare ethics. Unless considered otherwise, these features should be clearly identified in the abstract, the introduction and the conclusion. Similarly, the structure of the paper could be improved; the discourse is often fragmented and contents could be arranged differently. The case scenarios need to be revised. The reference list is broad but some key references may be missing (cf. specific comments).   

Specific Comments

Title - the title is concise but arguably not entirely clear. The title should emphasise the scope of the paper (i.e. the role of ethics in informing veterinary responsibilities towards animal welfare).

Additional wording has been added to help clarify, as suggested.

Simple summary – This section reflects poorly the content of the paper and should be revised.

Revised, as suggested

2. Ethics in Veterinary Medicine – This chapter is about veterinary ethics education. Shouldn’t the word ‘education/teaching/training’ be part of the title?

Added “education”, as suggested.

 2.1 Why is ethics taught? This section requires restructuring and further insight. The first paragraph (lines 43-46) is not clearly connected with the following paragraphs and something is missing in between. Why do the authors think that veterinary ethics teaching in the 1970s-80s was about professional etiquette and not about euthanasia?

This is based on the writing of Bernard Rollin, including his textbook “An Introduction to Veterinary Medical Ethics” and an article in JAVMA from 1978. It is also supported by comments from reviewer 1.

What has changed since?

An expansion of ethics teaching – added in sentences which make this flow better

The section then examines recent advances in veterinary ethics education in Europe, Canada and the US. Would you consider including Australasia for a truly global perspective?

Note that we have already included ethics teaching in Australasia in section 2.2

This geographic sequence could be repeated in the next section. Furthermore, research from Main et al (lines 75-79), is about how ethics is taught and should be part of the next section.

Moved, as suggested.

Finally, better use of reference 8 - one of the few attempts to explore why veterinary ethics is taught - is advised.

This section has been expanded to include major themes from interviews of educators. We have also included another reference which makes the point about moral stress.

2.2 How is veterinary ethics taught?

Line 92 -… (including ethical frameworks and approaches), laws….

Revised, as suggested

Line 93 – Ref [8]. This reference seems wrong. The authors probably meant https://doi.org/10.1136/vr.102553. Furthermore, this reference could be used to guide the remaining of the section and help justify the choice of the case scenarios (cf. below).

Corrected, this was the intended reference.

Line 116 - --- such as In Practice’s Everyday Ethics …

Corrected, as suggested

Lines 132-143 – Again, could codes of conduct from jurisdictions other than Canada be used to inform this section?

We have added additional references from other jurisdictions into Section 3 to address this comment.

Line 149 onwards – I think that by now the reader has been guided towards a next step of ethical enquiry and that a new section on something like ‘veterinary ethical decision-making’/’ethical responsibilities’ should start here.

New subheading added and other subheadings renumbered

Paragraph in lines 156-162 should be moved further down.  

Revised, as suggested

Line 345 onwards – The authors chose 7(6?) examples of prominent veterinary ethical issues with animal welfare implications. Several issues arise: What guided the choice of these case scenarios? Are they supposed to be genuine ethical dilemmas or just challenges? Why focusing in cases of animal welfare and not in cases of professionalism (which are also part of vet ethics)? Why choosing two exotic and one wild species and not presenting horses, pigs or poultry? Moreover, further efforts should be made to connect the case scenarios with the previous sections on vet ethics education (in terms of e.g. ethical frameworks, professional guidelines, regulations…). For example, virtue ethics, ethics of care and codes of conduct have not informed any of the cases. You could probably apply different frameworks to different cases as to illustrate their rationale. Alternatively, some of the information provided in the previous sections (especially in 4. Ethics and animal welfare in veterinary practice) could be fed into the case scenarios.

The cases were selected from those within our specific realm of practice – unfortunately, we do not see commercial swine or poultry on a regular basis. The case scenarios have been renumbered and expanded to better connect them with the above sections and improve flow, as suggested.

Example 1 – I think that this example lacks insight. A recent investigation from Ireland unravelled the complexity of issues involved when deciding to transport and slaughter injured livestock https://doi.org/10.1186/s13620-017-0102-0, including the role of client pressure on clinical decision-making, the availability of on-farm emergency slaughter and defective regulations.

Thank-you for your comment. We agree that transportation decisions can be challenging and have added the reference indicated as well as referring back to the ethics of care. All of the cases that we discuss are complex and the vet and client involved always have a choice.

Example 2 – Line 412 – References should be added pointing toward the incompetence of exotic pet owners (e.g. https://doi.org/10.1016/j.jveb.2017.07.005).

Reference added, as suggested, and additional information has been added linking back to Table 1.

Example 3 – The authors should allude do the concept of that ‘X-raying pockets’ (i.e., trying to predict how much the client will want to pay) may prevent the client from having a full awareness of the available treatment options (c.f. https://doi.org/10.1136/vr.158.2.62 and Ref. 31)

Revised, as suggested.

Example 4 – The concept of autonomy should be discussed in light of the concept of informed consent (c.f. https://doi.org/10.1007/s41055-017-0016-2). What should have been done differently?

Revised, as suggested. The objective of the case scenarios is not to explain a unique corrective solution but to promote self-reflection and to analyze the consequences of the referring veterinarians’ actions.

Example 6 - ???

Thank-you we have renumbered the cases.

Example 7 – This is a strange scenario and not entirely clear. Did the vet overlook the extent of the burns in his first visit? When did he forget the remaining cows? I think this is more a case of professional negligence than truly an ethical challenge. Couldn’t this case be replaced by an alternative case on broiler welfare? (e.g. vet detected lame broilers, prescribed new feed, one week later the broilers couldn’t stand up; decided for euthanasia, despite substantial economic loss…).

This case scenario could be an example only of negligence, but the ethical challenge is based on keeping alive a highly genetically valuable specimen and because the veterinarian was conflicted between client and animal distress. We have added some additional information to make this clearer.

Conclusion – In its present form the conclusion is insufficient. I suggest that the information provided in the Simple Summary should be moved to the conclusion. I am not totally in agreement to separate between ‘animal welfare issues’ and ‘ethical dilemmas’ (again, are the case scenarios one, the other, or both?). Please elaborate on the statement “all veterinarians require a strong ethical framework”; it seems to imply that we just need to pick one from Table 1 and we’re done! The authors should point toward the future, including recommendations in terms of educational needs in animal welfare science, ethics and law for veterinary students /veterinarians. Regarding the case scenarios, the authors should clarify that there are arguably several possible approaches to successfully resolve them and that the presented suggestions are not the only solutions.

The conclusion has been significantly revised, as suggested.

Reviewer 2 Report

All of my comments may be shown to both authors and editors.

Author Response

This reviewer provided a helpful discussion about the history of veterinary ethics as a discipline, contextualising the need for a paper like this. Thank you.

Reviewer 3 Report

Overall, this is an excellent paper on a very important topic, and the paper is very timely. There is currently a societal movement regarding speaking up, and the title and topic of this paper focus on the veterinary role in speaking up on behalf of animals. The use of case examples to frame the discussion on ethical decision making will be useful to veterinary educators and practitioners. 

Suggestions:

This paper discusses how veterinary ethics is taught in veterinary curricula, with mention that animal welfare has become a hot topic at veterinary continuing education events. However, this article does not provide a clear way for practicing veterinarians to improve their understanding and knowledge of ethical decision making and communication skills to address the issues presented later in the paper. What should the average veterinarian do if they see themselves in any of the case examples? This article might make them feel guilty - what action can they take to do better for the next patient?

Minor edits:

Line 43: Add a semicolon after "aspects of professionalism".

Line 65: NAVMEC is referenced - consider also referencing the OIE day 1 competencies for veterinarians or the OIE core curriculum (http://www.oie.int/Veterinary_Education_Core_Curriculum.pdf). 

Line ~100: Consider mentioning the Animal Welfare Judging and Assessment Contest as an opportunity for veterinary students (and a few private practitioners) to practice animal welfare assessments and communication (www.awjac.org). 

Table 1: This table needs reformatting - for example, "utilitarianism" should all fit on one line. Consider adding horizontal lines between each of the ethical frameworks to make it easier to read.

Line 146: Add a comma after "refer".

Line 155: Add quotations after "owner or animal?".

Author Response

Reviewer 3

Suggestions:

This paper discusses how veterinary ethics is taught in veterinary curricula, with mention that animal welfare has become a hot topic at veterinary continuing education events. However, this article does not provide a clear way for practicing veterinarians to improve their understanding and knowledge of ethical decision making and communication skills to address the issues presented later in the paper. What should the average veterinarian do if they see themselves in any of the case examples? This article might make them feel guilty - what action can they take to do better for the next patient?

Thank-you for your comment. The tone of the paper has been revised, Section 5 has been added to address a means by which practitioners can improve their ethical reasoning in addition to other material and references included throughout the MS.

Minor edits:

Line 43: Add a semicolon after "aspects of professionalism".

Corrected, as suggested.

Line 65: NAVMEC is referenced - consider also referencing the OIE day 1 competencies for veterinarians or the OIE core curriculum (http://www.oie.int/Veterinary_Education_Core_Curriculum.pdf). 

Added, as suggested.

Line ~100: Consider mentioning the Animal Welfare Judging and Assessment Contest as an opportunity for veterinary students (and a few private practitioners) to practice animal welfare assessments and communication (www.awjac.org). 

Revised to include, as suggested.

Table 1: This table needs reformatting - for example, "utilitarianism" should all fit on one line. Consider adding horizontal lines between each of the ethical frameworks to make it easier to read.

Revised to landscape format to address comment.

Line 146: Add a comma after "refer".

Corrected, as suggested.

Line 155: Add quotations after "owner or animal?".

Corrected, as suggested.

Reviewer 4 Report

This is a well written and thoughtful manuscript. I believe it emphasizes some very important thoughts about the need for teaching ethics to veterinary students and encouraging the use of ethical constructs in veterinary communications and deliberation. The use of case studies in the piece is useful for illustrating the points emphasized in the rest of the text.

I have several recommendations for change in the manuscript.

I believe the "Simple Summary" at the beginning should be rewritten. As it stands it sounds very judgmental and is off-putting. On first read it led me to believe the manuscript might be filled with judgmental appraisals of what veterinarians do right and wrong - particularly the second sentence. I was pleased to see the tone in the remainder of the article was deliberative and balanced and well referenced. I believe the authors can construct a better summary of the article that welcomes the readership.

I liked the attempt to formulate table 1. But it doesn't work. As it is currently formatted the material in the columns runs vertically such that it is difficult to see which descriptions of attributes in columns 2, 3 and 4 describe the category in column 1. I think the authors would do better to make this table into a hierarchy of bullets that run across the page with divisions between each of the 'ethical frameworks'. I do believe the material is useful, but needs to be restructured. Perhaps with the type of hierarchical presentation you would find in a powerpoint slide.

I think the authors need to doublecheck their references. For example on page 8, first paragraph, references 42 and 43 are cited, but I don't think these references describe the biological, vs affective, vs natural living views of welfare. And reference #1 and #30 appear to be the same reference?

With these modifications, I would recommend the manuscript for publication.

Author Response

Reviewer 4

I have several recommendations for change in the manuscript.

I believe the "Simple Summary" at the beginning should be rewritten. As it stands it sounds very judgmental and is off-putting. On first read it led me to believe the manuscript might be filled with judgmental appraisals of what veterinarians do right and wrong - particularly the second sentence. I was pleased to see the tone in the remainder of the article was deliberative and balanced and well referenced. I believe the authors can construct a better summary of the article that welcomes the readership.

Significantly revised, as suggested.

I liked the attempt to formulate table 1. But it doesn't work. As it is currently formatted the material in the columns runs vertically such that it is difficult to see which descriptions of attributes in columns 2, 3 and 4 describe the category in column 1. I think the authors would do better to make this table into a hierarchy of bullets that run across the page with divisions between each of the 'ethical frameworks'. I do believe the material is useful, but needs to be restructured. Perhaps with the type of hierarchical presentation you would find in a powerpoint slide.

We have revised the table to a landscape format to address this comment.

I think the authors need to doublecheck their references. For example on page 8, first paragraph, references 42 and 43 are cited, but I don't think these references describe the biological, vs affective, vs natural living views of welfare.

Citations have been corrected throughout.

And reference #1 and #30 appear to be the same reference?

Thank-you – please note that reference #1 is a citation of Rollin’s work and reference #30 is specific to a book chapter.

Round 2

Reviewer 1 Report

The authors have successfully dealt with all the comments and suggestions. Well done!

Only a few details remain:

Add reference 5 to the references in line 102.

Line 104 – Reference 5 instead of 4.

References 29 and 30 should be merged.

References 1 and 36 should be merged. You can include page numbers in the text e.g. [1, pp.10-14]

Lines 183-184  “…Royal College of Veterinary Surgeons’ Code of Professional Conduct outlines veterinarians’ responsibilities to…” instead of “Royal College of Veterinary Surgeon’s Code of Professional Conduct for Veterinary Surgeon’s outlines veterinarian’s responsibilities to…”

Line 456 – Reference 70 instead of 69.

Line 463 – References 71-72 instead of 70-72.

Line 463 – There is little of virtue ethics in the described action. I suggest that you try to allude to the intentions of the vet (e.g.  … the referring vet could have expressed genuine concern and have asked…)

Reference 64 – Responding instead of responsing